# Template-jumping prime editing enables large insertion and exon rewriting in vivo

Chunwei Zheng [1,5], Bin Liu [1,5], Xiaolong Dong[1], Nicholas Gaston [1], Erik J. Sontheimer [1,2,3] ✉ & Wen Xue [1,2,3,4] ✉

Targeted insertion of large DNA fragments holds promise for genome engineering and gene therapy. Prime editing (PE) effectively inserts short (<50 bp) sequences. Employing paired prime editing guide RNAs (pegRNAs) has enabled PE to better mediate relatively large insertions in vitro, but the efficiency of larger insertions (>400 bp) remains low and in vivo application has not been demonstrated. Inspired by the efficient genomic insertion mechanism of retrotransposons, we develop a template-jumping (TJ) PE approach for the insertion of large DNA fragments using a single pegRNA. TJ-pegRNA harbors the insertion sequence as well as two primer binding sites (PBSs), with one PBS matching a nicking sgRNA site. TJ-PE precisely inserts 200 bp and 500 bp fragments with up to 50.5 and 11.4% efficiency, respectively, and enables GFP (~800 bp) insertion and expression in cells. We transcribe split circular TJ-petRNA in vitro via a permuted group I catalytic intron for non-viral delivery in cells. Finally, we demonstrate that TJ-PE can rewrite an exon in the liver of tyrosinemia I mice to reverse the disease phenotype. TJ-PE has the potential to insert large DNA fragments without double-stranded DNA breaks and facilitate mutation hotspot exon rewriting in vivo.

Prime editing is a powerful CRISPR-based genome editing approach that enables flexible genomic alterations, including all possible base substitutions, small genomic insertions, and small genomic deletions[1–4]. PE usually consists of a Cas9 nickase–reverse transcriptase (RT) fusion protein and prime editing guide RNA (pegRNA). The use of two pegRNAs (e.g., TwinPE and GRAND editing[5,6]) was recently shown to achieve the insertion of larger DNA fragments in cells, but the efficiency of large insertions (>400 bp) remains low. Furthermore, PE shows modest efficiencies in vivo[7–10]. Neither TwinPE nor GRAND editing has been applied in vivo. For many genetic disorders, the disease-related gene can harbor diverse mutations that cause a pathogenic phenotype[11–13]. Developing individual PE therapies for each pathogenic variant would be expensive and time-consuming. However, rewriting a mutation hotspot exon could provide a broadly

applicable treatment strategy for genetically diverse patients. Such an approach would require PE to achieve efficient large DNA insertions.

We hypothesized that the genomic insertion mechanism of retrotransposons could be harnessed to facilitate the targeted insertion of large DNA fragments. Non-long terminal repeat (non-LTR) retrotransposons are abundant in mammalian genomes[14,15]. For example, L1 retrotransposons, up to 6 kb in length, account for ~ 17% of the human genome. Retrotransposons replicate their genetic information into chromosomal DNA through target-primed reverse transcription (TPRT)[14,15]. In TPRT, the endonuclease nicks one strand of the target site to generate a DNA flap that anneals to the 3' end of retrotransposon mRNA, which works as a primer to enable first-strand synthesis by RT (Fig. 1a, left). The second strand nick occurs downstream or upstream of the first strand nick to generate a second primer

[1]RNA Therapeutics Institute, University of Massachusetts Chan Medical School, Worcester, MA 01605, USA. [2]Department of Molecular Medicine, University of Massachusetts Chan Medical School, Worcester, MA 01605, USA. [3]Li Weibo Institute for Rare Diseases Research, University of Massachusetts Chan Medical School, Worcester, MA 01605, USA. [4]Department of Molecular, Cell and Cancer Biology, University of Massachusetts Chan Medical School, Worcester, MA 01605, USA. [5]These authors contributed equally: Chunwei Zheng, Bin Liu. ✉e-mail: erik.sontheimer@umassmed.edu; wen.xue@umassmed.edu

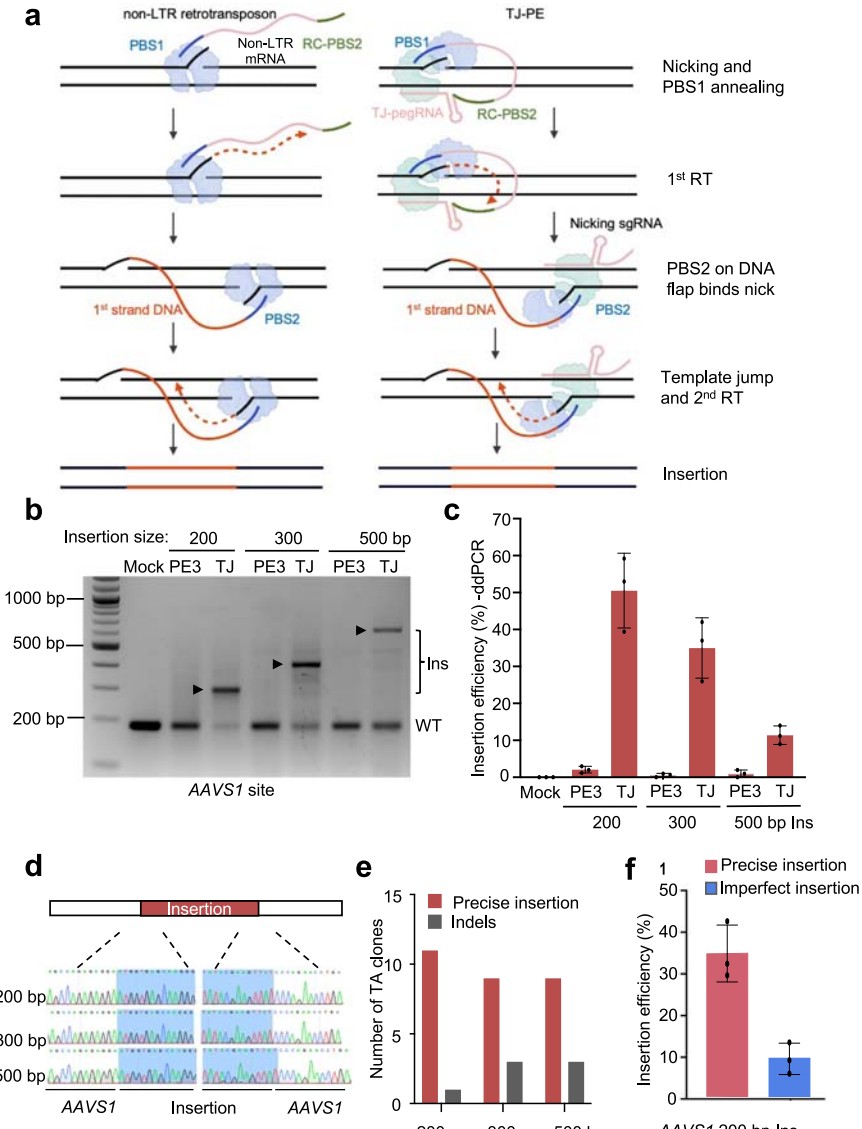

**Fig. 1 | Retrotransposon-like template jump prime editing (TJ-PE) mediates large genomic insertions. a** Schematics of non-LTR (left) and TJ-PE (right) mechanisms. TJ-pegRNA: template jump prime editing guide RNA; PBS1: primer binding site 1; RC-PBS2: reverse complement sequence of PBS2. **b** Insertion of DNA fragments with PE3 control or TJ-PE at *AAVS1* site. HEK293T cells were transfected with PE2, nicking sgRNA, and either TJ-pegRNA (TJ-PE) or control pegRNA (PE3). PCR using primers flanking *AAVS1* detected amplicons of 200, 300, and 500-bp insertions with a deletion of 90 bp at the *AAVS1* locus. Insertion bands of expected size are denoted with arrows. Ins: insertion, WT: wild-type. **c** Insertion efficiency at *AAVS1* locus measured by ddPCR. Results were obtained from three independent experiments, shown as mean ± s.d. **d** The insertion bands were gel purified. Sanger sequencing shows accurate insertions. **e** Precise insertion was confirmed by TA cloning and Sanger sequencing of 12 individual clones. **f** Absolute total precise insertion efficiency of a 200-bp DNA fragment at the *AAVS1* locus was quantified by deep sequencing. Results were obtained from three independent experiments (*n* = 3), shown as mean ± s.d.

---

that anneals to the 3' end of the newly-synthesized minus strand, which becomes the new template for synthesis of the plus strand[14,15].

Here, we develop a template-jumping prime editor (TJ-PE) (Fig. 1a, right) to mimic TPRT and enable precise insertions of large DNA fragments (up to 800 bp) at endogenous sites. We achieve insertion efficiencies of up to 50.5% for 200 bp and 11.4% for 500 bp in cells. In a mouse model of tyrosinemia, TJ-PE is able to rewrite a mutated exon of the *Fah* gene in the liver and rescue the disease phenotype.

## Results

### Retrotransposon-like TJ-PE enables large insertion

To mimic TPRT, we designed a TJ-pegRNA and nicking sgRNA as shown in Fig. 1a (right). The 3' extension of TJ-pegRNA contains an insertion sequence (red), primer binding site 1 (PBS, blue), and a reverse complement sequence of PBS2 (RC-PBS2, green). After PE and TJ-pegRNA

nick the top DNA strand, the resulting DNA flap hybridizes to the PBS1 sequence, and the RT domain of PE synthesizes the first DNA strand. The newly synthesized DNA contains the desired insertion fragment and a PBS2 sequence at the 3' end. PBS2 is designed to hybridize to the second nicked site generated by PE and nicking sgRNA to initiate the template jump and second strand synthesis.

We designed TJ-pegRNA to insert 200-, 300-, or 500-bp DNA fragments into the *AAVS1* locus. TJ-pegRNAs contained a trimmed evopreQ1 (tevopreQ1) motif at the 3' end, as this feature has been shown to enhance pegRNA stability and improve prime editing efficiency[16]. The TJ-pegRNA and nicking sgRNA sites are 90 bp apart, resulting in deletion of a 90-bp genomic fragment with the desired fragment insertion. Following transfection of HEK293T cells with TJ-pegRNA, nicking sgRNA, and PE, PCR amplification of the target region showed a band of the predicted insertion size at the *AAVS1* site

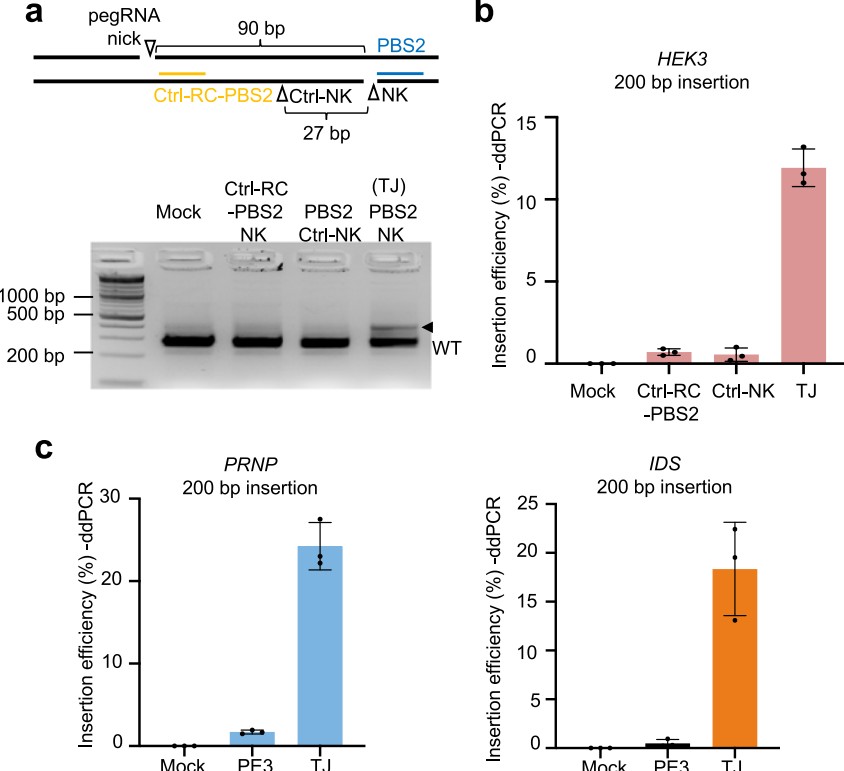

**Fig. 2 | TJ-PE mediates insertions at multiple genomic loci. a** Insertion of a 200-bp DNA fragment at *HEK3* locus by TJ-PE. HEK293T cells were transfected with PE2, nicking sgRNA, and either pegRNA with a control RC-PBS2 (ctrl-RC-PBS2) or a control nicking sgRNA (ctrl-NK) as controls. The insertion band of the predicted size was observed following TJ-PE treatment but not controls (arrow). **b** Insertion

efficiency at *HEK3* measured by ddPCR. **c** Insertion of DNA fragments with PE3 control (pegRNA with a control RC-PBS2 sequence) or TJ-PE at *PRNP* (left) and *IDS* (right) loci. Insertion efficiency was measured by ddPCR. Results were obtained from three independent experiments (*n* = 3), shown as mean ± s.d.

(Fig. 1b). Control pegRNAs were designed to produce a PBS2 complementary to a site 46 bp upstream of the nicking sgRNA site (termed PE3 control). The PE3 control showed no clear band of the predicted insertion length (Fig. 1b), suggesting that base pairing of PBS2 to the DNA flap at the nicking sgRNA site is essential for effective insertion. Droplet digital polymerase chain reaction (ddPCR) using primers spanning the junction sequence of the insertion showed that the average insertion efficiency of TJ-PE was 50.5% for the 200-bp insertion, 35.1% for the 300-bp insertion, and 11.4% for the 500-bp insertion. The insertion efficiency of the PE3 control was 19- to 35-fold lower for the 200-, 300-, and 500-bp insertions (2.1%, 1.0%, and 0.6%, respectively; Fig. 1c) compared to TJ-PE.

To determine the accuracy of DNA fragment insertion, we gel purified the PCR bands of the expected insertion sizes (Supplementary Fig. 1a). Sanger sequencing showed that these fragments were completely aligned with the expected inserted sequences (Fig. 1d). TA cloning of individual clones with Sanger sequencing estimated precise insertion rates in the expected insertion band to be 91.7%, 75.0%, and 75.0% for 200-bp, 300-bp, and 500-bp insertions, respectively (Fig. 1e and Supplementary Fig. 1b-d). The remaining TA clones harbored imperfect insertion or insertion with point mutations (Supplementary Fig. 1b-d). To determine the absolute total precise insertion efficiency, we sequenced PCR products via deep sequencing. TJ-PE generated an accurate insertion frequency of 34.3% for the 200-bp insertion at the *AAVS1* locus, while the efficiency of imprecise insertion was 9.5% (Fig. 1f).

Next, we compared TJ-pegRNA and PE3 at multiple endogenous sites. We inserted a 200-bp DNA fragment at the endogenous *HEK3* locus in HEK293T cells. The TJ-pegRNA and nicking sgRNA sites are 90 bp apart, resulting in a deletion of the 90-bp DNA fragment coupled

to a 200-bp insertion. As a pegRNA control, we designed a pegRNA with an RC-PBS2 matching a sequence directly 3′ of the pegRNA nicking site (ctrl-PBS2). As a nicking sgRNA control, we designed a nicking sgRNA (ctrl-NK) to target 27 bp upstream of the site complementary to PBS2 (Fig. 2a, top) to generate a 63-bp deletion with a 200-bp insertion. Using gel electrophoresis and ddPCR, we determined that the insertion efficiency of TJ-pegRNA was significantly higher than ctrl-PBS2 and ctrl-NK groups (11.9%, 0.7%, and 0.6%, respectively; Fig. 2b). Additionally, no insertion band was detected at the *HEK3* locus when the control nicking sgRNA was designed to nick at the same position as nicking sgRNA but on the opposite strand, indicating that PBS2 hybridizing to the second nicked site to initiate template jump and second-strand synthesis is essential for TJ-PE (Supplementary Fig. 2a).

Next, we assessed TJ-PE-mediated insertion of a 200-bp fragment with concomitant 72-bp or 70-bp deletions at the endogenous *PRNP* or *IDS* loci, respectively. PegRNAs were designed to produce a PBS2 complementary to a sequence directly 3′ of the pegRNA nicking site (termed PE3 control). We found that TJ-PE was 14-fold more efficient than PE3 at the *PRNP* site (24.2% versus 1.7%, respectively) and 37-fold more efficient than PE3 at the *IDS* site (18.4% versus 0.5%, respectively, Fig. 2c and Supplementary Fig. 2b, c). To explore the broader application of the TJ-PE strategy across commonly used cell lines, we tested the ability of TJ-PE to insert a 200-bp fragment at *PRNP*, *CCR5*, and *IDS* sites in A549 and U-2 OS cells. TJ-PE enabled insertion (3.3–8.3%) across loci and cell lines (Supplementary Fig. 2d, e).

To determine whether PBS2 length impacts insertion efficiency, we designed TJ-pegRNA with different RC-PBS2 lengths (13 bp, 17 bp, and 35 bp) and measured their abilities to insert a 200-bp fragment at the *HEK3* locus. All TJ-pegRNAs supported similar insertion efficiencies

(11.0%, 12.3%, and 9.3%; Supplementary Fig. 3a, b). We also compared the efficiency of inserting a GFP fragment sequence without or with a hairpin structure (introduced by LoxP). Insertion efficiencies were similar between groups, suggesting LoxP did not impede the activity of reverse transcriptase, possibly due to the presence of RNA helicases which could unwind hairpin structures in cells. (Supplementary Fig. 3c).

PegRNAs are prone to misfolding due to inevitable base pairing between the PBS and spacer sequence, which could potentially contribute to lower insertion efficiency[9,17]. To stabilize the pegRNA and prevent misfolding, we designed a nicking-TJ-pegRNA (NK-TJ-pegRNA) containing a PBS1 sequence that first hybridizes to the DNA flap generated by the nicking sgRNA (Supplementary Fig. 4a). However, the NK-TJ-pegRNA did not increase insertion efficiency at the *AAVS1* site as compared with TJ-pegRNA [62.5 versus 59.2% (for 200-bp insertion) and 41.4 % versus 42.2% (for 300-bp insertion), respectively] (Supplementary Fig. 4b, c).

We next investigated whether tethering the PBS1 sequence of TJ-pegRNA to the PE fusion protein – via an MS2 coat protein (MCP) – could improve TJ-pegRNA stability and enhance insertion efficiency (Supplementary Fig. 5a). To accomplish this, we inserted the MS2 aptamer sequence at the 3′ end of TJ-pegRNA instead of tevopreQ1

motif, and MCP into the PE fusion protein sequence. To determine whether MCP placement affects results, we tested different MCP fusion sites in the PE protein: at the N terminus, C terminus, or between the nCas9 and RT segments of PE (Supplementary Fig. 5b). We found that, regardless of configuration, TJ-pegRNA tethered to PE-MCP protein did not increase insertion efficiency at the *HEK3* locus compared to untethered TJ-pegRNA and PE (Supplementary Fig. 5c). GRAND editing employs a pair of pegRNAs[6], which can efficiently generate the insertion of DNA fragments up to 400 bp (Supplementary Fig. 6a). We compared the efficiency of TJ-PE and GRAND editing in inserting a 200-bp, 400-bp or 500-bp DNA fragment at multiple endogenous sites. Our results showed that TJ-PE and GRAND editing mediate similar insertion rates (Supplementary Fig. 6b-d).

## GFP reporter repair and functional gene insertion

To investigate whether TJ-PE could generate large in-frame insertions to restore gene expression, we employed a HEK293T traffic light reporter/multi-Cas variant 1 (TLR-MCV1) cell line that contains a disrupted green fluorescent protein (GFP) sequence with a 39-bp sequence insertion and an mCherry sequence separated by a T2A sequence[18]. The mCherry sequence is out of frame with the disrupted GFP sequence, preventing mCherry expression (Fig. 3a). Precise repair

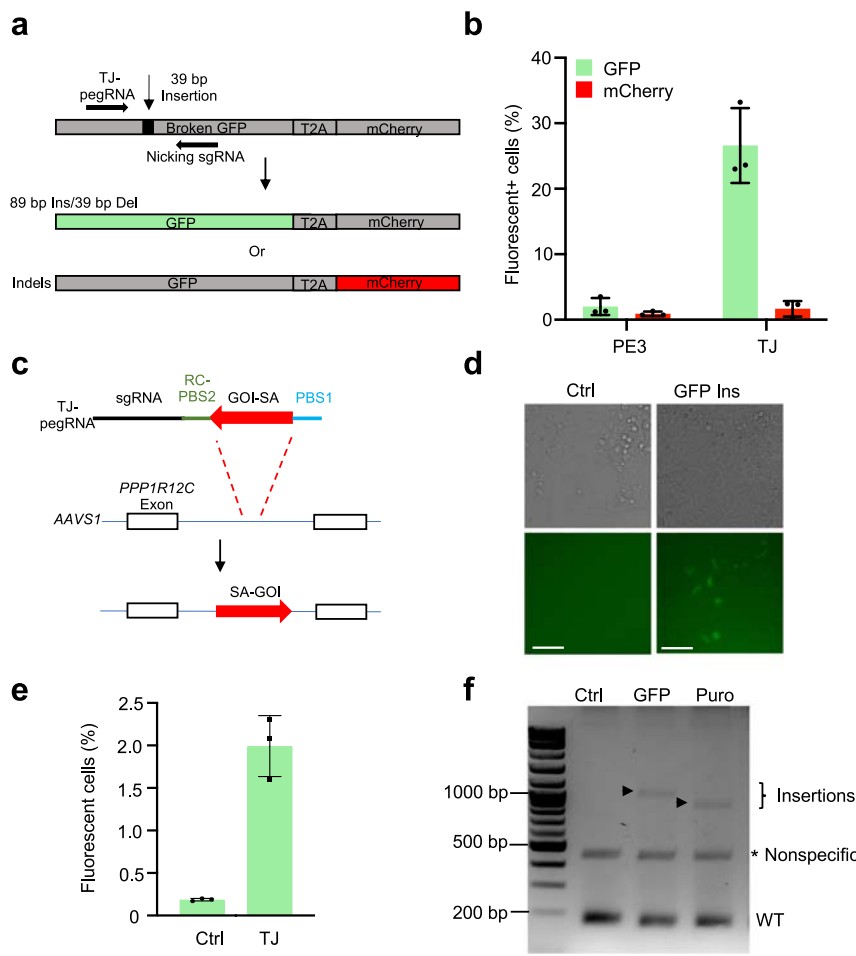

**Fig. 3 | TJ-PE mediates GFP reporter and functional gene insertion. a** A diagram of the TLR-MCV1 reporter line. Inserting an 89-bp sequence to replace the 39-bp non-functional sequence results in GFP expression. Indels result in mCherry expression. Del: deletion. **b** PE3 control and TJ-PE were tested in the TLR-MCV1 reporter line, and flow cytometry was used to determine the percentage of fluorescent cells. Results were obtained from three independent experiments, shown as mean ± s.d. **c** Schematics of TJ-pegRNA and targeting strategy for inserting SA-GOI at *AAVS1* locus. SA: splice acceptor; GOI: gene of interest. **d** Bright field and

fluorescence images of HEK293T cells 4 days after transfection with PE, TJ-pegRNA, and nicking sgRNA. HEK293T cells transfected with PE plasmid only served as a control (ctrl). Experiments were done two times, and one is shown. Scale bar: 100 μm. **e** Efficiency of SA-GFP insertion measured by flow cytometry. Results obtained from three independent experiments, shown as mean ± s.d. **f** Agarose gel of PCR amplicons showing SA-GFP and SA-Puro insertion. Puro: puromycin. The insertion bands of expected sizes are indicated with arrow. The nonspecific bands are indicated with asterisk. Experiments were done two times, and one is shown.

of the disrupted sequence will enable GFP expression; indels that shift into the +1 reading frame will induce mCherry expression. We treated TLR-MCV1 cells with PE, TJ-pegRNA, and nicking sgRNA designed to precisely insert an 89-bp codon-optimized fragment and concomitantly delete the 39-bp disruption sequence. A pegRNA designed to insert a 73-bp codon-optimized fragment and concomitantly delete the 39-bp disruption sequence was used as the PE3 control. TJ-PE led to a 13-fold increase in the level of precise 89-bp insertion compared to control (26.6% versus 2.0%, respectively, Fig. 3b and Supplementary Fig. 7a); the indel efficiency was also higher in the TJ-PE-treated group than in the control group (1.7% versus 0.9%, respectively, Fig. 3b and Supplementary Fig. 7a). These data demonstrate that TJ-PE can repair genomic coding regions through precise, large, in-frame insertions.

To explore TJ-PE limits with respect to insertion size, we designed TJ-pegRNA to insert either splice acceptor (SA)-GFP (833 bp) or SA-Puro (709 bp) at the *AAVS1* locus after deleting a 90-bp DNA fragment (Fig. 3c). Using fluorescence microscopy, we observed EGFP[+] cells in the TJ-PE-treated group (Fig. 3d). Flow cytometry analysis showed that the EGFP[+] cell efficiency was 2.0% (Fig. 3e and Supplementary Fig. 7b). The control group (plasmid encoding PE protein only) showed minimal EGFP-positive cells (0.2%). After confirming insertions were the expected sizes (Fig. 3f), we purified the insertion bands and found that these fragments were precisely inserted using Sanger sequencing (Supplementary Fig. 7c). These data demonstrate that TJ-PE can mediate functional gene insertion at the *AAVS1* site.

## Split circular TJ-petRNA enables large insertion for non-viral delivery

Non-viral (RNA-based) delivery of gene editors has considerable therapeutic potential for a wide range of diseases due to its many advantages, including ease of scale-up, transient expression, lack of immune response, and minimum off-target effects[19–21]. However, pegRNA needs to be quite long to generate large insertions (e.g., 226-nt TJ-pegRNA is needed for a 100-bp insertion), making RNA synthesis complex. Long pegRNAs can be transcribed in vitro, but this does not allow for the addition of chemical modifications to improve pegRNA stability.

Previous reports suggest that in vitro transcribed circular RNAs exhibit not only higher stability, but also lower immunogenicity, compared to unmodified linear RNA[20]. To develop an RNA-encoded TJ-pegRNA system, we split TJ-pegRNA into an sgRNA and a prime editing template RNA (petRNA) carrying an RTT-PBS sequence and an MS2 stem-loop aptamer, as previously reported[9]. The MS2-RTT-PBS was designed to form a circular RNA via a permuted group I catalytic intron in vitro[22,23] (Fig. 4a and Supplementary Fig. 8a). Split circular TJ-petRNA was tethered to the MCP-RT fusion protein by the MS2 aptamer (Fig. 4b). To test circularization efficiency, we treated the transcribed RNA with RNase R (digests linear, but not circular RNA) and RNase H. We observed a circularization efficiency of >90% (Fig. 4c). Circular RNAs were enriched using RNase R and electroporated into HEK293T cells along with sgRNA, nicking sgRNA, and mRNAs encoding nCas9 and MCP-RT. Deep sequencing showed that split circular

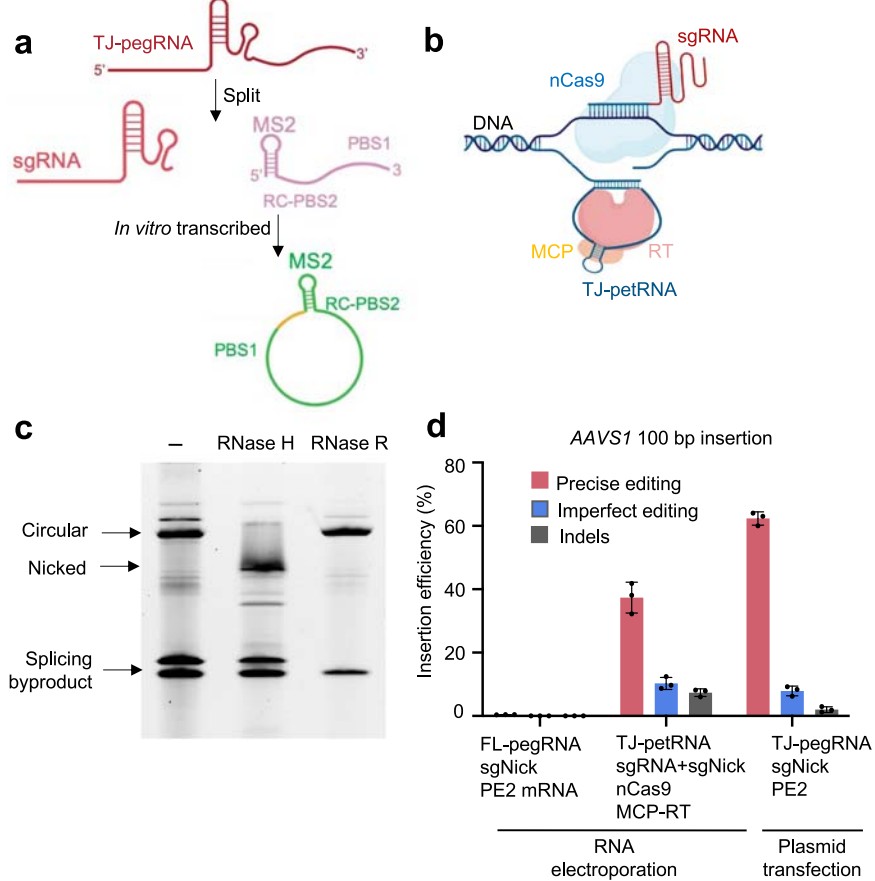

**Fig. 4 | In vitro transcribed split circular TJ-petRNA enables large insertion. a** Illustration of split circular TJ-petRNA. The prime editing template RNA (petRNA) sequence carrying an RTT-PBS sequence and an MS2 stem-loop aptamer, and circularized via a permuted group I catalytic intron. Yellow: circularization sequence. **b** Model of split circular petRNA function in PE. **c** Urea polyacrylamide gel showing split circular TJ-petRNA after splicing, RNase H, and RNase R digestion. Linear, but not circular, RNA is digested by RNase R. Experiments were done two times, and one is shown. **d** Editing efficiency of split circular TJ-petRNA at the *AAVS1* locus. Synthesized sgRNAs and in vitro transcribed split circular petRNA were co-transfected with nCas9 and MCP-RT mRNA in HEK293T cells. For comparison purposes, HEK293T cells were transfected with PE2, TJ-pegRNA, and nicking sgRNA plasmids. FL-pegRNA: in vitro transcribed full-length TJ-pegRNA. Results were obtained from three independent experiments, shown as mean ± s.d.

TJ-petRNA mediates 37.6% insertion at the *AAVS1* locus (Fig. 4d and Supplementary Fig. 8b). We in vitro transcribed full-length linear TJ-pegRNA without chemical modification, which supported low efficiency (0.4%) likely due to the instability of unmodified RNA. As a control, transfected TJ-pegRNA plasmid generates an accurate insertion frequency of 62.3% (Fig. 4d). Long TJ-pegRNA (>150nt, as well as paired PE pegRNA) is difficult or impossible to produce via current solid-phase RNA synthesis with modified nucleotides. Our findings highlight TJ-petRNA using the split circular template as a promising alternative to overcome the limitations of long RNA synthesis for non-viral delivery. Our results demonstrate that in vitro transcribed, circular MS2-containing petRNA can be coupled with TJ-PE to enable DNA fragment insertion, increasing the feasibility of using an RNA-encoded TJ-PE system to achieve large DNA insertion in vivo.

## TJ-PE mediated recoding of the *Fah* exon 8 locus in the Tyrosinemia I mouse model

Tyrosinemia I is an autosomal recessive disorder characterized by hepatocyte toxin accumulation and liver damage. Tyrosinemia I is caused by loss-of-function mutations in the fumarylacetoacetate hydrolase (*FAH*) gene[24]. Tyrosinemia I mice harbor a G•C to A•T point mutation in the last nucleotide of exon 8 in the *Fah* gene[25], resulting in exon 8 skipping and loss of functional FAH protein (Fig. 5a).

Tyrosinemia I mice need to be treated with 2-(2-nitro-4-tri-fluoromethylbenzoyl)−1,3-cyclohexanedione (NTBC) supplemented water to maintain body weight and survive. Multiple types of mutations in exon 8 have been reported in patients[26].

Replacing the mutant *Fah* exon sequence with a synonymous DNA fragment would correct any combination of mutations in the exon. This exon rewriting strategy has the potential to correct multiple pathogenic mutations using a single template. We engineered TJ-pegRNA and nicking sgRNA targeting the genomic region across exon 8 (Fig. 5b). TJ-pegRNA harbors the correction "G" and multiple synonymous mutations. PE2, TJ-pegRNAs, and nicking sgRNA (Nicking sgRNA-1) plasmids were delivered to the livers of mice via hydrodynamic injection. Two weeks later, immunohistochemistry analysis of TJ-PE-treated liver sections revealed 0.1% of hepatocytes were FAH positive (Supplementary Fig. 9a, b). Since hepatocytes with corrected FAH protein gain a growth advantage[25], we removed the NTBC water and observed that *Fah*-mutant mice treated with saline control rapidly lost 15% of body weight, while TJ-PE showed body weight rescued 45 days after NTBC withdrawal (Fig. 5c). We observed widespread FAH-positive cell clusters in TJ-PE-treated mouse livers by immunohistochemistry 35 days after NTBC withdrawal (Supplementary Fig. 9c). The efficiency of precise replacement was confirmed via deep sequencing two months after NTBC withdrawal

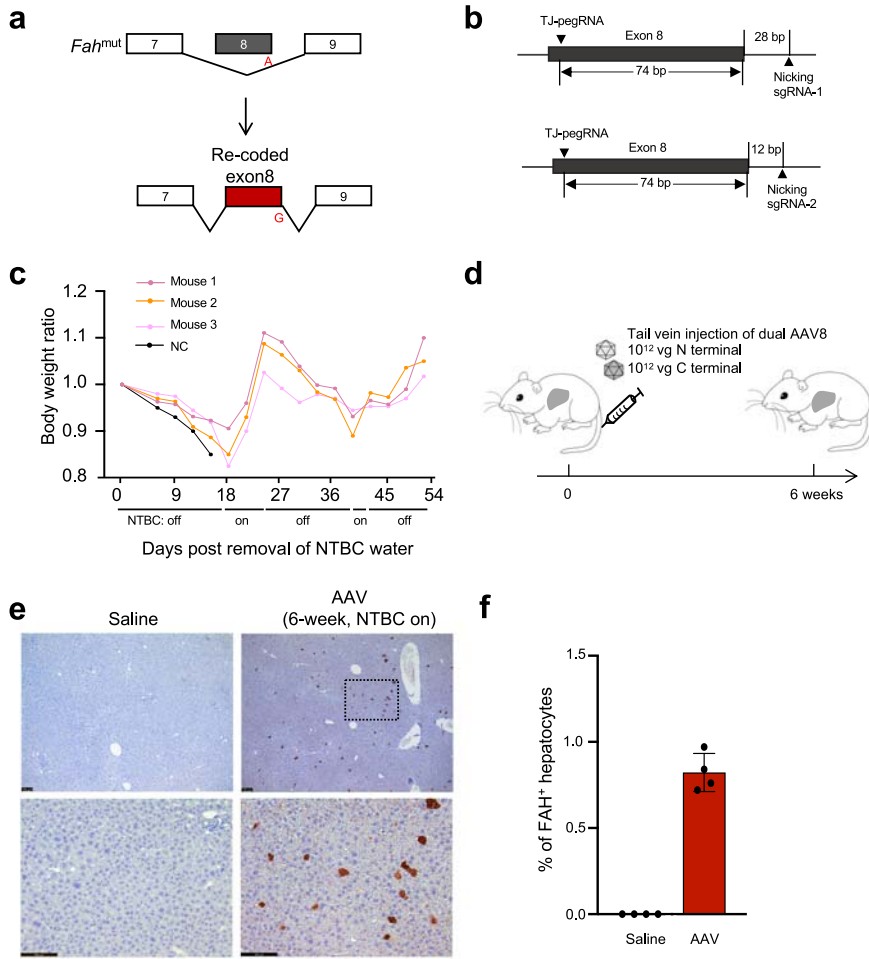

**Fig. 5 | TJ-PE rewrites a correction exon in mouse liver. a** Diagram of *Fah* splicing before and after correction by TJ-PE. **b** A diagram of the TJ-PE strategy at *Fah* locus. **c** TJ-PE treatment rescues body weight after NTBC withdrawal. The body weight ratio is normalized to day 0 of NTBC withdrawal. NC: treated with Saline. **d** Schematic of the split-intein dual AAV8 system and tail vein injection experimental timeline. Four-week-old tyrosinemia I mice were injected with a total of $2 \times 10^{12}$ vg AAV8. **e** Representative FAH IHC images. Scale bars, 100 μm. Mice treated with saline were used as negative controls. The lower panel of AAV is a high-magnification view (box with black line). **f** Quantification of FAH+ hepatocytes by IHC six weeks after AAV injection. Data represent mean ± SD. (*n* = 4 mice for each group).

(average 2.5%, Supplementary Fig. 9d). We also observed sequencing reads with partial synonymous mutations and/or the correction "G" incorporated (Supplementary Fig. 9e), which may be due to the RTT being highly homologous to the genomic sequence. To reduce imperfect editing and improve precise editing rate, we optimized the RTT to avoid microhomology with the genomic sequence and used a new nicking sgRNA which is closer to the rewritten exon to include less intron sequence (Fig. 5b, Nicking sgRNA-2). We delivered this optimized TJ-PE using a dual-AAV8 split-intein system to *Fah*-mutant mice that were then kept on NTBC-supplemented water for 6 weeks to prevent the expansion of *Fah*-corrected cells (Fig. 5d, Supplementary Fig 9f). Immunohistochemistry revealed up to 1.0% of hepatocytes stained positive for the FAH protein in AAV-treated animals (Fig. 5e, f). Overall, our data demonstrate the potential of using TJ-PE in vivo to rewrite an exon with hotspot mutations and suggest the need to further optimize pegRNA to increase editing efficiency.

## Discussion

In this study, we engineered a retrotransposon-inspired TJ-pegRNA to enable precise insertion of large (up to 800 bp) DNA fragments at endogenous genomic sites. We applied TJ-PE to rewrite a mutation hotspot exon in vivo, demonstrating the potential of using TJ-PE to develop a broadly applicable strategy to correct multiple pathogenic variants.

TJ-PE uses only one pegRNA compared to paired PE using two long pegRNAs, reducing the complexity of RNA synthesis and delivery. For plasmid-based experiments, we used the U6 promoter to transcribe long TJ-pegRNA in cells. Yet the long TJ-pegRNA (as well as paired PE pegRNA) is difficult or impossible to produce via solid-phase RNA synthesis for stabilization with modified nucleotides. To address this issue, we uncoupled the RTT-PBS from the sgRNA and transcribed split circular TJ-petRNA in vitro via a permuted group I catalytic intron (Fig. 4). This approach has the potential to overcome the limitations of long RNA synthesis for non-viral delivery.

We also expanded the size limit of PE-based genomic insertions and demonstrated the insertion of an 800 bp GFP cassette. This size range can support the insertion of small functional genes or reporters at endogenous promoters. The modest rate of 800 bp insertion efficiency is possibly due to the instability of the newly synthesized DNA strand prior to binding the nicked site. Stabilization of the newly synthesized DNA by fusion to an ssDNA-binding protein domain (ssDBD)[27,28] might, therefore, enhance the efficiency of TJ-PE. TJ-PE efficiency and accuracy might also be restricted by asynchrony between PBS2 synthesis and generation of the second nicked site, which dictates the timing and efficiency of PBS2 annealing to the resulting DNA flap for second-strand synthesis. Optimizing the nicking sgRNA and using an induced system may help PBS2 bind to the nicking sgRNA target site. Besides precise insertions, TJ-PE introduced imperfect insertion and indels (Figs. 1e and 4d). Future studies will delineate how these key steps limit the efficiency and precision of TJ-PE and therefore find a way to improve it.

Finally, we demonstrated that TJ-PE could rewrite an exon to replace a mutation-harboring exon in a mouse model of genetic disease. This approach can be applied to correct multiple types of hot spot mutations within an exon or a genomic region. In summary, TJ-PE has the potential to expand the scope of genome editing to enable the insertion of large DNA sequences.

## Methods
### Ethical statement
All animal experiments were approved by the Institutional Animal Care and Use Committee (IACUC) at University of Massachusetts Chan Medical School (PROTO202000051).

### Plasmid construction
Plasmids expressing sgRNA were constructed by ligation of annealed oligonucleotides into pmd264 vector (BfuAI digested) that has been described previously in ref. 29. To generate pegRNA plasmids, gBlocks gene fragments (spacer, scaffold, and 3' extension sequences) were synthesized by Integrated DNA Technologies, and subsequently cloned into a BfuAI/EcoRI-digested pmd264 vector by Gibson assembly. Sequences of sgRNA and pegRNA are listed in Supplementary Table 1. Plasmids used for in vitro experiments were purified using Miniprep kits (Qiagen). Plasmids were purified using a Maxiprep kit (Qiagen) including the endotoxin removal step for in vivo experiments.

### Cell culture, transfection and genomic DNA isolation
HEK293T cells, U-2 OS cells and A549 cells acquired from ATCC were maintained in Dulbecco's Modified Eagle's Medium (DMEM) supplemented with 10% (v/v) fetal bovine serum (Gibco) and 1% (v/v) Penicillin/Streptomycin (Gibco). Cells were cultured at 37 °C with 5% $CO_2$.

HEK293T cells were seeded on 12-well plates overnight at 100,000 cells per well. One microgram PE2, 500 ng pegRNA, and 500 ng nicking sgRNA were transfected using Lipofectamine 3000 (Invitrogen). Cells were collected 4 days after transfection, lysed with 100 μL Quick extraction buffer (Epicenter), and incubated on a thermocycler at 65 °C for 15 min and 98 °C for 5 min. Sequences of primers used for genomic DNA amplification are listed in Supplementary Table 2.

### Droplet Digital PCR (ddPCR)
ddPCR was used to quantify the amplicon containing the insertion fragment or insertion-genome junction in comparison to a reference amplicon. Briefly, gDNA was added to a reaction containing ddPCR Supermix (no dUTP, Bio-Rad), the primers (900 nM) and the probes (250 nM). Droplets were generated using a QX200 Manual Droplet Generator (Bio-Rad). PCR reactions were carried out as follows: 95 °C for 10 min, 36 cycles of 94 °C for 30 s and 58 °C for 1 min, 98 °C for 10 min, and 4 °C holds. Droplets were read using a QX200 Droplet Reader (Bio-Rad) and analyzed using QuantaSoft (Bio-Rad). The results were concatenated and loaded into GraphPad Prism 8.4 for data visualization. Sequences of probes are listed in Supplementary Table 3.

### Flow cytometry analysis
Flow cytometry analysis was performed on day 4 after transfection. Reporter cells were collected after PBS washing and trypsin digestion and resuspended in PBS with 2% FBS for flow cytometry analysis (MACSQuant VYB). Data were analyzed by FlowJo 10.0 software.

### In vitro transcription
The transcription of split circular TJ-petRNA was performed as previously described[22]. The template was synthesized by Integrated DNA Technologies and amplified via PCR. Split circular TJ-petRNA was generated at 37 °C for 4 h using a HiScribe T7 High-Yield RNA Kit (New England Biolabs) according to the manufacturer's protocol. After DNase I digest, 0.8ul 100 mM GTP was added to 1 reaction, 55 °C for 15 min. The RNA was then purified using a Monarch RNA Cleanup kit (New England Biolabs).

### Nucleofection
The Neon electroporation system was used for electroporation. Briefly, 1 μg of each mRNA, 100 pmol of sgRNA, 100 pmol of nicking sgRNA, and 30 pmol split circular TJ-petRNA were electroporated into $5 \times 10^4$ HEK293T cells. One microgram of each mRNA, 100 pmol of pegRNA, and 100 pmol of nicking sgRNA was electroporated as control group. HEK293T cells were electroporated using the following electroporation parameters: 1150 V, 20 ms, two pulses.

## Deep sequencing and data analysis

Sequencing library preparation was performed as previously described[1]. Briefly, for the first round of PCR, the primers containing Illumina forward and reverse adapters (listed in Supplementary Table 4) were used for amplifying the genomic sites of interest from 100 ng genomic DNA using Phusion Hot Start II PCR Master Mix. PCR 1 reactions were carried out as follows: 98 °C for 10 s, then 20 cycles of 98 °C for 1 s, 58 °C for 5 s, and 72 °C for 6 s, followed by a final 72 °C extension for 2 min. A secondary PCR reaction were performed to add a unique Illumina barcode to each sample from 1 μL unpurified PCR 1 product. PCR 2 reactions were carried out as follows: 98 °C for 10 s, then 20 cycles of 98 °C for 1 s, 60 °C for 5 s, and 72 °C for 8 s, followed by a final 72 °C extension for 2 min. PCR 2 products were purified by gel purification using the QIAquick Gel Extraction Kit (Qiagen). DNA concentration was measured by Qubit dsDNA HS Assay Kit (Thermo Fisher Scientific). The library was sequenced on an Illumina MiniSeq instrument following the manufacturer's protocols. Sequencing reads were demultiplexed using bcl2fastq (Illumina). To quantify the frequency of precise editing and indels, CRISPResso2[30] was run in HDR mode with "plot_window_size" = 65, "default_min_aln_score" = 60, 'min_average_read_quality' = 30. The indel efficiency was calculated as 100% - precise insertion - WT, and then normalize to a blank group.

## Animal studies

Mice were housed in cages with ad libitum access to food and water, in rooms with controlled temperature and illumination (12 h/12 h light-dark cycle). All plasmids used for hydrodynamic tail-vein injection were prepared using EndoFree Plasmid Maxi kit (Qiagen). *Fah* mutant B6/Tyr- mice were kept on 10 mg/L NTBC water. Thirty micrograms of PE2, 15 μg TJ-pegRNA, and 15 μg nicking sgRNA were injected into 6-to 8-week-old (*n* = 6, male and female) *Fah* mutant B6/Tyr- mice via the tail vein in 5–7 s. Saline was injected in the control group (*n* = 6, male and female). NTBC-supplemented water was replaced with normal water 7–14 days after injection, and mouse weight was measured daily. AAV8 was resuspended in 200 μl of 0.9% NaCl and administered to 4-week-old female *Fah* mutant B6/Tyr- mice (*n* = 4, male and female) via tail vein injection. Saline was injected in the control group (*n* = 4, male and female). Different groups were allocated in a randomized manner and investigators were blinded to the allocation of different groups. The primary antibodies anti-FAH (ab83770, Abcam Inc, dilution: 1:400) were used.

## AAV production

Low-passage HEK293T cells were transfected with AAV genome, pHelper, and Rep/Cap plasmids using PEI. After three days, the cells were dislodged and transferred to 50 mL Falcon tubes. For AAV purification, 1/10 Volume of pure chloroform was added and shaken vigorously at 37 °C for 1 h. NaCl was added to a final concentration of 1 M, followed by centrifugation at 20,000 g at 4 °C for 15 min. The supernatant was gently collected and PEG8000 (Sigma) was added for virus precipitation. The pellet was resuspended in DPBS containing MgCl2 and Benzonase (Sigma), and incubated at 37 °C for 45 min. Chloroform was added to remove protein and the aqueous layer was ultrafiltered through 100 kDa MWCO columns (Millipore). The virus titer was quantified via qPCR[31].

## Statistics and reproducibility

At least two or three replicates were analyzed in each independent experiment to ensure the experimental data were reliable. Data are expressed as mean ± s.d. Statistical analyses were performed with GraphPad Prism (version 8.0, GraphPad Software).

## Reporting summary

Further information on research design is available in the Nature Portfolio Reporting Summary linked to this article.

## Data availability

The raw sequencing data have been deposited in the NCBI BioProject database under accession code PRJNA973981. Source data are provided as a Source Data file. Source data are provided with this paper.

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

## Acknowledgements
We thank H. Yin and P. Zamore for helpful discussions and E. Haberlin for editing the manuscript. We thank Y. Liu in the UMass Chan Morphology Core for support. This work was supported by grants from the National Institutes of Health (DP2HL137167, P01HL158506, and UH3HL147367) and the Cystic Fibrosis Foundation. X.D. and E.J.S. were supported in part by the National Institutes of Health (R01GM115911 and UG3TR002668), Leducq Foundation and the Rett Syndrome Research Trust.

## Author contributions
C.Z., B.L., E.J.S., and W.X. designed the research. C.Z. performed experiments, analyzed data, and wrote the manuscript. B.L., X.D., and N.G. performed experiments. All authors reviewed, edited, and approved the manuscript.

## Competing interests
E.J.S. is a co-founder and Scientific Advisory Board member of Intellia Therapeutics and a Scientific Advisory Board member at Tessera Therapeutics. The University of Massachusetts Chan Medical School has filed a patent application on TJ-PE in this work (inventors: C.Z., B.L., X.D., E.J.S. and W.X, patent filed/pending). The remaining authors declare no competing interests.
