## [Peer Review File · Nature Communications]

Reviewers' Comments:

Reviewer #1:

Remarks to the Author:

In the manuscript "Template-jumping prime editing enables large insertion and exon rewriting in vivo", Zehng et al. describe a novel strategy based on prime editing to insert large DNA fragments in cells and in animals.

The method consists on using a modified pegRNA harboring two PBS flanking the Reverse Transcriptase Template (RTT). One of the two PBS maps immediately upstream of the nick induced by the pegRNA, as in conventional prime editing, while the other is designed to anneal with the region immediately 5' to a nick generated by a separate sgRNA on the opposite strand. This design promotes template switching by the RT, in a process reminiscent of the mechanisms of integration of retrotransposons.

The authors demonstrate the efficiency and accuracy of this novel approach in HEK293 cells, inserting templates of up to ~800 base-pairs with efficiency ranging from ~50% to 2%. They also demonstrate the therapeutic potential of this in a mouse model of tyrosinemia. Finally, they propose a clever strategy to generate a split circular TJ-pegRNA in which the two PBS and the RTT are separated from the sgRNA portion and encoded as a circular—and therefore more stable—RNA.

This work is aimed at addressing a major current limitation of prime editing (and of genome editing in general): the relatively low efficiency with which large DNA fragments can be inserted into the genome. Overcoming this limitation in the context of prime-editing would have major implications for genome engineering in general and for gene therapy in particular.

The manuscript is well written, the experiments are generally well controlled and described in sufficient detail. The relevant scientific literature is cited, including recently proposed alternative strategies based on paired pegRNAs, and the statistical analysis is, as far as I can judge, appropriate.

The strategy proposed is innovative and elegant, and it is certainly amenable of further improvements. As such, it may be of general interest to the broad readership of Nature Communications.

There are a few issues that this reviewer would like to see addressed by the authors:

- 1) Although the authors claim that the TJ-PE method outperforms alternative strategies recently proposed, a direct side-by-side comparison of TJ-PE, twinPE, and GRAND editing on the same locus and using the same cleavage sites/RTT would be particularly useful to support this claim.
- 2) For the data shown in Figure 1, accuracy of insertion is probably overestimated since the authors purified and sequenced only bands of the expected size. Judging from panel 1b, there are several additional bands, some substantially smaller, that are only present in the TJ samples and are likely the results of imperfect insertions. The authors should submit the entire PCR product for high throughput sequencing to obtain a more accurate picture.
- 3) Figure 4. To better evaluate the TJ-petRNA method it would be important to provide a side-by-side comparison with TJ-pegRNA using the same insert and the same nicking sites. Unless I misunderstood, the insert used for the experiments described in figure 4 is different from the insert used for the experiments described in figure 1.
- 4) For the in vivo experiments (fig 5) it is unclear to me how big was the fragment inserted. How much of the flanking introns are also included?
- 5) For figure 5, the repair efficiency was determined at 2 months after NTBC withdrawal. Since repaired liver cells have a strong selective advantage after NTBC withdrawal in this model, this is probably greatly overstating the true repair efficiency. The authors should quantify repair efficiency at an earlier time point after hydrodynamic tail vein injection, before NTBC withdrawal,

to obtain a more accurate readout of the true efficiency of in vivo TJ-PE.

Reviewer #2:

Remarks to the Author:

The authors used a mouse model of tyrosinemia type 1, caused splice mutation in the Fah gene. Correction of this mutation will result in a selective advantage of gene corrected hepatocytes and the formation of Fah positive nodules.

I have several comments on this part of the paper:

- 1) The number of corrected animals and controls is minimal, too low to be meaningful.
- 2) The frequency of editing needs to be clarified. The authors claim a precise replacement frequency of 3.1%. What does this number mean exactly. It appears to be derived from harvesting livers after selection by NTBC withdrawal. Does this number then mean that 3.1% of hepatocytes were gene corrected by their strategy or that the final percentage of liver with corrected hepatocytes was 3.1% AFTER selection. If the number is derived from the post-selection livers, then the true frequency of precise editing is much lower than 3.1%. What was the total percentage of liver repopulation at the end of the experiment? This is not apparent from the data presented. Previous publications have used the frequency of Fah+ nodules along with a correction factor for nodule size to estimate the initial frequency of gene corrected cells. The authors should apply this method.
- 3) Fig. 5C: show Fah IHC at low magnification, so that the frequency of gene corrected nodules is visible.
- 4) Clarify whether the weight curves shown in Fig. 5B were obtained from animals cycled on and off NTBC. If this is the case, indicate the periods of re-initiating NTBC on the graph. The up and down of the weight almost certainly reflects NTBC cycling. It appears that the mice got an NTBC break on day 18.

Reviewer #3:

Remarks to the Author:

In this manuscript Zheng and Liu et al. have developed a new Prime Editing strategy called Template-jumping Prime Editing (TJ-PE), which is inspired by the mechanism of non-LTR retrotransposition. This strategy allows for the insertion of large DNA fragments into cells. The authors demonstrated the effectiveness of TJ-PE in HEK293T cells by inserting fragments up to 800bp, which are usually too large insertions for standard prime editors. They also used TJ-PE to correct a genetic disorder called Tyrosinemia I in a mouse model. Although the efficiency was low and the fragment size was relatively small (100bp), the authors provide evidence of prime editing in mice, making it a valuable achievement.

Long fragments insertion is a significant limitation of prime editing and a new technology allowing this could be of broad interest. However, the report is preliminary, and the advantage of TJ-PE compared to existing methods such as GRAND is not clearly demonstrated. One advantage of TJ-PE is that the system only requires one pegRNA (although a second gRNA is still required) as opposed to two pegRNAs for TwinPE and GRAND, but this in itself is not an incentive large enough to adapt TJ-PE broadly. A more significant advantage of TJ-PE could be efficient insertion of fragments larger than 400bp, but this is not strongly demonstrated in the current manuscript.

Major points:

- The authors compare TJ-PE to PE3 throughout the paper. While using PE3 is acceptable as a negative control, it is not a benchmark prime editing approach to insert long DNA fragments as it not expected to be able to insert fragments >40bp. The comparison of TJ-PE to PE3 in figures 1, 2 and 3 is thus not entirely fair.
- GRAND or TwinPE would be much more suitable controls and could establish TJ-PE as a preferred

method for larger fragment insertion. The authors do briefly mention these two approaches as inefficient, however, GRAND was demonstrated to be efficient for insertions up to 300bp. In fact, GRAND was reported to install 250bp insertions with up to 30% editing efficiency and it would be important to perform head to head comparison at least for the three target sites reported in the Figure 2 of this manuscript.

- The authors don't prove that TJ-PE is working by Target-Primed Reverse Transcription and it is not simply a method to promote standard prime editing. I suggest to include a control where the nicking gRNA nicks at the same position but on the opposite strand (same strand used for first reverse transcription event).
- Expanding the number of target sites tested would strengthen the current manuscript. To demonstrate the broader usefulness of this strategy for cell engineering, the authors need to conduct experiments in cell lines other than HEK293T.
- It is uncertain from the presented data if there are any rules beyond optimal sequences that can be efficiently inserted by TJ-PE. To address this, it would be beneficial to provide data comparing insertions of standard sequences (used in the current version of the manuscript) to the same sequences containing hairpins, such as LoxP sites. This comparison is significant because it has been demonstrated that hairpins can effectively halt the activity of reverse transcriptase and could prevent TJ-PE mediated insertions
- While it does appear that the main strength of TJ-PE could be its ability to install insertions larger than 300bp (this is where the efficiency of the dual pegRNA approaches drops dramatically), the current manuscript only reports very few data regarding larger fragments (500bp and ~800bp at AAVS1 at 11% and 2% respectively). To solidify this part of the study, more comprehensive data on larger fragments (≥ 400 bp) across different target sites and inclusion of GRAND for comparison would be essential to demonstrate TJ-PE robustness and superiority over existing methods.
- In vivo data is based on hydrodynamic injection of plasmid DNA. This approach is not suitable for use in humans. I believe the authors need to use a more standard method of delivery, such as AAV or PE mRNA/ template RNA encapsulation in LNP.
- The authors have reported that the average efficiency of the TJ-PE treatment In vivo is 3.1%. However, they also mention that fah cells that have been corrected with TJ-PE tend to have a growth advantage. This may mean that the actual efficiency of the treatment is being overestimated. To get a clearer understanding of the effectiveness of TJ-PE and any potential limitations of the system, it would be helpful to examine the efficiency of the editing at an earlier stage, such as day 14.

Minor points:

- Figure 5: The size of the fragment that is being rewritten in vivo is not stated anywhere in the manuscript (or cannot be found easily) and the reader is forced to count nucleotides in the printed sequence. Please indicate the size in the main text or in Figure 5. A schematic of TJ-PE strategy at fah site would be useful.
- It is not clear the identity of other events that are not precise insertions and the mechanism of integration. An example is in Fig3f where a more prominent insertion is present but the authors don't clarify the mechanism of insertion.

General Response to Reviewers

We thank the reviewers for their constructive feedback, which has allowed us to greatly improve our manuscript. We have made significant revisions to the paper after careful consideration of all comments and suggestions. These changes include performing the following additional experiments to further investigate the robustness of the TJ-PE system and to compare it directly with GRAND editing:

- We have performed head-to-head comparisons between TJ-PE and GRAND editing at multiple target sites (**Supplementary Fig 6**).
- We have delivered TJ-PE using dual-AAVs to rewrite a mutation hotspot exon in the liver of tyrosinemia I mice (**Fig. 5d-f**).
- We tested TJ-PE editing at a total of six genomic sites (**Fig. 1b; Fig. 2b, c; Fig. 5f and Supplementary Fig. 2d, e**). For three endogenous sites, we tested across two cell lines (**Supplementary Fig. 2d, e**).
- For TJ-PE testing at one endogenous site, we included a new control nicking sgRNA designed to nick at the same position as our original nicking sgRNA control but on the opposite strand (**Supplementary Fig. 2a**).
- We performed high throughput sequencing to calculate the absolute accurate insertion frequencies of 100-bp and 200-bp insertions at the *AAVS1* locus (**Fig. 1f and 4d**).
- We have compared editing efficiencies for TJ-petRNA versus TJ-pegRNA using the same insert and the same nicking site (**Fig. 4d**).
- We have compared TJ-PE mediated insertion of the same sequence with or without a hairpin (**Supplementary Fig. 3c**).

Below is a point-by-point response to specific comments from each reviewer:

Reviewer #1

In the manuscript "Template-jumping prime editing enables large insertion and exon rewriting in vivo", Zheng et al. describe a novel strategy based on prime editing to insert large DNA fragments in cells and in animals.

The method consists on using a modified pegRNA harboring two PBS flanking the Reverse Transcriptase Template (RTT). One of the two PBS maps immediately upstream of the nick induced by the pegRNA, as in conventional prime editing, while the other is designed to anneal with the region immediately 5' to a nick generated by a separate sgRNA on the opposite strand. This design promotes template switching by the RT, in a process reminiscent of the mechanisms of integration of retrotransposons.

The authors demonstrate the efficiency and accuracy of this novel approach in HEK293 cells, inserting templates of up to ~800 base-pairs with efficiency ranging from ~50% to 2%.

They also demonstrate the therapeutic potential of this in a mouse model of tyrosinemia. Finally, they propose a clever strategy to generate a split circular TJ-pegRNA in which the two PBS and the RTT are separated from the sgRNA portion and encoded as a circular—and therefore more stable—RNA.

This work is aimed at addressing a major current limitation of prime editing (and of genome editing in general): the relatively low efficiency with which large DNA fragments can be inserted into the genome. Overcoming this limitation in the context of prime-editing would have major implications for genome engineering in general and for gene therapy in particular.

The manuscript is well written, the experiments are generally well controlled and described in sufficient detail. The relevant scientific literature is cited, including recently proposed alternative strategies based on paired pegRNAs, and the statistical analysis is, as far as I can judge, appropriate.

The strategy proposed is innovative and elegant, and it is certainly amenable of further improvements. As such, it may be of general interest to the broad readership of Nature Communications.

We thank the reviewer for the positive feedback and constructive comments.

There are a few issues that this reviewer would like to see addressed by the authors:

1) Although the authors claim that the TJ-PE method outperforms alternative strategies recently proposed, a direct side-by-side comparison of TJ-PE, twinPE, and GRAND editing on the same locus and using the same cleavage sites/RTT would be particularly useful to support this claim.

We thank the reviewer for this suggestion. We have updated our manuscript to include a comparison of the efficiency of TJ-PE and GRAND editing in inserting a 200-bp, 400-bp, or 500-bp DNA fragment at multiple endogenous sites. Our results showed that TJ-PE and GRAND editing mediate similar insertion rates (**Supplementary Fig. 6b-d**). However, it should be noted that TJ-PE uses only one long pegRNA, while GRAND editing employs a pair of pegRNAs (**Supplementary Fig. 6a**). This unique feature of TJ-PE makes it possible to generate large insertions using RNA-based delivery (TJ-petRNA). Our findings suggest it may be possible to exploit this unique feature of TJ-PE without sacrificing editing efficiency.

2) For the data shown in Figure 1, accuracy of insertion is probably overestimated since the authors purified and sequenced only bands of the expected size. Judging from panel 1b, there are several additional bands, some substantially smaller, that are only present in the TJ samples and are likely the results of imperfect insertions. The authors should submit the entire PCR product for high throughput sequencing to obtain a more accurate picture.

Whereas **Fig. 1e** shows the accuracy efficiency in the expected insertion PCR band, the absolute total insertion efficiency is shown in **Fig. 1c**. We used ddPCR, which has been widely used to map genomic insertions, to calculate absolute insertion efficiency in this experiment. Because of the 100- to 400-bp size difference between the unedited and edited sequence, PCR-based next-generation sequencing of total genomic DNA will lead to preferential bias of the smaller WT amplicon thereby resulting in an underestimation of accurate editing rates.

Nevertheless, we have performed high-throughput sequencing to calculate accurate insertion frequencies for 100-bp and 200-bp insertions at the *AAVS1* locus. Accurate insertion frequency in total genomic DNA was 34.3% for the 200-bp insertion and 62.3% for the 100-bp insertion (**Fig. 1f and 4d**).

3) Figure 4. To better evaluate the TJ-petRNA method it would be important to provide a side-by-side comparison with TJ-pegRNA using the same insert and the same nicking sites. Unless I misunderstood, the insert used for the experiments described in figure 4 is different from the insert used for the experiments described in figure 1.

Fig. 4 uses electroporation of circular TJ-petRNA as the insertion template for a 100bp insertion.

As suggested, we have compared TJ-petRNA and TJ-pegRNA using the same insert and the same nicking site. This is not a direct side-by-side comparison, however, as we used plasmid transfection (pegRNA) versus electroporation of mRNA and split template (petRNA). We detected 62.3% insertion for TJ-pegRNA (plasmid) and 37.4% insertion for TJ-petRNA (electroporation). We have included these data as **Fig. 4d**. Long TJ-pegRNA (>150nt, as well as paired PE pegRNA) is difficult or impossible to produce via solid-phase RNA synthesis for stabilization with modified nucleotides. Our findings highlight TJ-petRNA as a promising alternative to overcome the limitations of long RNA synthesis for non-viral delivery.

4) For the in vivo experiments (fig 5) it is unclear to me how big was the fragment inserted. How much of the flanking introns are also included?

It was 74 bp exon and 27 bp intron edited. The size of the fragment inserted is shown in **Fig. 5b** (upper schematic).

5) For figure 5, the repair efficiency was determined at 2 months after NTBC withdrawal. Since repaired liver cells have a strong selective advantage after NTBC withdrawal in this model, this is probably greatly overstating the true repair efficiency. The authors should quantify repair efficiency at an earlier time point after hydrodynamic tail vein injection, before NTBC withdrawal, to obtain a more accurate readout of the true efficiency of in vivo TJ-PE.

We thank the reviewer for this suggestion. In this revision, we delivered TJ-PE plasmids via hydrodynamic injection to mice that were then kept on NTBC-supplemented water for 2 weeks (to prevent the expansion of *Fah*-corrected cells) then sacrificed. Immunohistochemistry (IHC) analysis of livers sections revealed an initial precise editing efficiency of ~0.1% (**Supplementary Fig. 9a, b**).

To improve precise editing and minimize imperfect editing (something we observed at 2 months post-NTBC withdrawal in our initial manuscript), we optimized the RTT sequence to avoid microhomology with the genomic sequence and used a new nicking sgRNA that is closer to the rewritten exon and includes less intron sequence (**Fig 5b**, lower schematic). We delivered TJ-PE using a dual-AAV8 split-intein system into *Fah*-mutant mice that

were then kept on NTBC-supplemented water for 6 weeks prior to sacrifice (**Fig. 5d**). IHC analysis showed up to 1.0% hepatocytes stained positive for the FAH protein (**Fig. 5e, f**).

Reviewer #2

The authors used a mouse model of tyrosinemia type 1, caused splice mutation in the *Fah* gene. Correction of this mutation will result in a selective advantage of gene corrected hepatocytes and the formation of *Fah* positive nodules.

I have several comments on this part of the paper:

1) The number of corrected animals and controls is minimal, too low to be meaningful.

In this revised manuscript, we carried out an additional experiment in *Fah*-mutant mice using TJ-PE with an optimized RTT sequence and a new nicking sgRNA designed to reduce imperfect editing and improve precise editing. We delivered the optimized TJ-PE (or saline) into *Fah*-mutant mice using a dual-AAV8 split-intein system. This experiment used n=4 mice/group (**Fig. 5e, f**).

2) The frequency of editing needs to be clarified. The authors claim a precise replacement frequency of 3.1%. What does this number mean exactly. It appears to be derived from harvesting livers after selection by NTBC withdrawal. Does this number then mean that 3.1% of hepatocytes were gene corrected by their strategy or that the final percentage of liver with corrected hepatocytes was 3.1% AFTER selection. If the number is derived from the post-selection livers, then the true frequency of precise editing is much lower than 3.1%.

What was the total percentage of liver repopulation at the end of the experiment? This is not apparent from the data presented.

Previous publications have used the frequency of *Fah*⁺ nodules along with a correction factor for nodule size to estimate the initial frequency of gene corrected cells. The authors should apply this method.

In our initial manuscript, we determined the efficiency of precise replacement via deep sequencing at two months post NTBC withdrawal. These data therefore reflect corrected hepatocytes after selection. To directly respond to the original question, we delivered TJ-PE plasmids via hydrodynamic injection to mice that were then kept on NTBC-supplemented water for 2 weeks (to prevent the expansion of *Fah*-corrected cells) prior to sacrifice. Subsequent IHC analysis of liver sections revealed an initial precise editing efficiency rate of ~0.1% (**Supplementary Fig. 9a, b**). To improve precise editing (and reduce imperfect editing), we then optimized the RTT sequence to avoid microhomology with the genomic sequence and used a new nicking sgRNA, which is closer to the rewritten exon to include less intron sequence. We delivered this optimized TJ-PE to *Fah*-mutant mice using a dual-AAV8 split-intein system, then kept mice on NTBC water for 6 weeks. IHC analysis of liver from AAV-treated mice showed up to 1.0% of hepatocytes stained positive for the FAH protein (**Fig. 5e, f**).

3) Fig. 5C: show *Fah* IHC at low magnification, so that the frequency of gene corrected nodules is visible. As suggested, FAH IHC at low magnification is now included in **Fig. 5e** and **Supplementary Fig. 9a**.

4) Clarify whether the weight curves shown in Fig. 5B were obtained from animals cycled on and off NTBC. If this is the case, indicate the periods of re-initiating NTBC on the graph. The up and down of the weight almost certainly reflects NTBC cycling. It appears that the mice got an NTBC break on day 18.

Two cycles of NTBC reintroduction were performed. The first occurred on D18-24 and the second on D39-42. Our revised manuscript indicates these NTBC cycles in **Fig. 5c**.

Reviewer #3 (Remarks to the Author):

In this manuscript Zheng and Liu et al. have developed a new Prime Editing strategy called Template-jumping Prime Editing (TJ-PE), which is inspired by the mechanism of non-LTR retrotransposition. This strategy allows for the insertion of large DNA fragments into cells. The authors demonstrated the effectiveness of TJ-PE in

HEK293T cells by inserting fragments up to 800bp, which are usually too large insertions for standard prime editors. They also used TJ-PE to correct a genetic disorder called Tyrosinemia I in a mouse model. Although the efficiency was low and the fragment size was relatively small (100bp), the authors provide evidence of prime editing in mice, making it a valuable achievement.

Long fragments insertion is a significant limitation of prime editing and a new technology allowing this could be of broad interest. However, the report is preliminary, and the advantage of TJ-PE compared to existing methods such as GRAND is not clearly demonstrated. One advantage of TJ-PE is that the system only requires one pegRNA (although a second gRNA is still required) as opposed to two pegRNAs for TwinPE and GRAND, but this in itself is not an incentive large enough to adapt TJ-PE broadly. A more significant advantage of TJ-PE could be efficient insertion of fragments larger than 400bp, but this is not strongly demonstrated in the current manuscript.

Major points:

- The authors compare TJ-PE to PE3 throughout the paper. While using PE3 is acceptable as a negative control, it is not a benchmark prime editing approach to insert long DNA fragments as it not expected to be able to insert fragments >40bp. The comparison of TJ-PE to PE3 in figures 1, 2 and 3 is thus not entirely fair.

Because TJ-PE uses a single pegRNA and one nicking sgRNA (similar to the traditional PE method), we think it is reasonable to compare it with PE3. However, we have also included experiments comparing TJ-PE and GRAND editing to better benchmark our method (see next response).

- GRAND or TwinPE would be much more suitable controls and could establish TJ-PE as a preferred method for larger fragment insertion. The authors do briefly mention these two approaches as inefficient, however, GRAND was demonstrated to be efficient for insertions up to 300bp. In fact, GRAND was reported to install 250bp insertions with up to 30% editing efficiency and it would be important to perform head to head comparison at least for the three target sites reported in the Figure 2 of this manuscript.

We thank the reviewer for this suggestion. In the revised manuscript, we have compared the efficiency of TJ-PE and GRAND editing in inserting a 200-bp fragment at multiple loci. We found TJ-PE and GRAND editing mediate similar rates of 200-bp insertion at *HEK3* (15.7% versus 24.5%), *IDS* (14.2% versus 10.9%), and *PRNP* (11.4% versus 17.9%). We have added these data as **Supplementary Fig. 6b,c**.

- The authors do not prove that TJ-PE is working by Target-Primed Reverse Transcription and it is not simply a method to promote standard prime editing. I suggest to include a control where the nicking gRNA nicks at the same position but on the opposite strand (same strand used for first reverse transcription event).

We thank the reviewer for this suggestion. In an experiment evaluating TJ-PE editing at the *HEK3* locus, we included a new control nicking sgRNA designed to nick at the same position as our nicking sgRNA but on the opposite strand. We did not detect the insertion band in this control group, indicating that hybridizing PBS2 to the second-nicked site to initiate template jump and second-strand synthesis is essential for TJ-PE. We included these data in **Supplementary Fig. 2a**.

- Expanding the number of target sites tested would strengthen the current manuscript. To demonstrate the broader usefulness of this strategy for cell engineering, the authors need to conduct experiments in cell lines other than HEK293T.

We tested a total of six genomic sites for TJ-PE (**Fig. 1b; Fig. 2b, c; Fig. 5f and Supplementary Fig. 2d, e**). For three endogenous sites, we tested across two different cell lines (**Supplementary Fig. 2d, e**).

- It is uncertain from the presented data if there are any rules beyond optimal sequences that can be efficiently inserted by TJ-PE. To address this, it would be beneficial to provide data comparing insertions of standard sequences (used in the current version of the manuscript) to the same sequences containing hairpins, such as **LoxP sites**. This comparison is significant because it has been demonstrated that hairpins can effectively halt the activity of reverse transcriptase and could prevent TJ-PE mediated insertions.

We thank the reviewer for raising this important question. We compared the efficiency of inserting a GFP fragment without or with a hairpin structure (introduced by LoxP). Insertion efficiencies were similar between groups, suggesting LoxP did not impede the activity of reverse transcriptase, possibly due to the presence of RNA helicases, which could unwind hairpins in cells (**Supplementary Fig. 3c, d**).

- While it does appear that the main strength of TJ-PE could be its ability to install insertions larger than 300bp (this is where the efficiency of the dual pegRNA approaches drops dramatically), the current manuscript only reports very few data regarding larger fragments (500bp and ~800bp at AAVS1 at 11% and 2% respectively). To solidify this part of the study, more comprehensive data on larger fragments (≥ 400 bp) across different target sites and inclusion of GRAND for comparison would be essential to demonstrate TJ-PE robustness and superiority over existing methods.

We compared the efficiency of TJ-PE and GRAND editing in inserting a 500-bp fragment at *AAVS1*, as well as a 400-bp fragment at the *CCR5*, *IDS*, and *PRNP* loci. We found that TJ-PE and GRAND editing mediate similar rates of insertion at *AAVS1* (12.3% versus 5.2%), *CCR5* (8.4% versus 3.4%), *IDS* (1.5% versus 1.8%) and *PRNP* (1.4% versus 1.8%). We have added these data as **Supplementary Fig. 6d**.

- In vivo data is based on hydrodynamic injection of plasmid DNA. This approach is not suitable for use in humans. I believe the authors need to use a more standard method of delivery, such as AAV or PE mRNA/ template RNA encapsulation in LNP.

As requested, we have included an AAV delivery experiment in the revised manuscript. Prior to encoding TJ-PE in AAVs, we optimized RTT further to avoid microhomology with the genomic sequence and used a new nicking sgRNA located closer to the rewritten exon, containing less intron sequence, with the goal of improving precise editing and reducing imperfect editing. We delivered TJ-PE using a dual-AAV8 split-intein system into *Fah*-mutant mice (n=4), which were then kept on NTBC-supplemented water for 6 weeks to prevent the expansion of *Fah*-corrected cells (**Fig. 5d**). Subsequent IHC analysis in livers of AAV-treated mice revealed up to 1.0% hepatocytes stained positive for the FAH protein (**Fig. 5e, f**).

- The authors have reported that the average efficiency of the TJ-PE treatment In vivo is 3.1%. However, they also mention that *fah* cells that have been corrected with TJ-PE tend to have a growth advantage. This may mean that the actual efficiency of the treatment is being overestimated. To get a clearer understanding of the effectiveness of TJ-PE and any potential limitations of the system, it would be helpful to examine the efficiency of the editing at an earlier stage, such as day 14.

We thank the review for this suggestion. In this revision, we delivered TJ-PE plasmids via hydrodynamic injection to mice that were then kept on NTBC-supplemented water for 2 weeks (to prevent the expansion of *Fah*-corrected cells) then sacrificed. Subsequent IHC analysis of liver sections revealed an initial precise editing efficiency of ~0.1% (**Supplementary Fig. 9a, b**).

Minor points:

- Figure 5: The size of the fragment that is being rewritten in vivo is not stated anywhere in the manuscript (or cannot be found easily) and the reader is forced to count nucleotides in the printed sequence. Please indicate the size in the main text or in Figure 5. A schematic of TJ-PE strategy at *fah* site would be useful.

A schematic of the TJ-PE strategy is shown in **Fig. 5b**.

- It is not clear the identity of other events that are not precise insertions and the mechanism of integration. An example is in Fig3f where a more prominent insertion is present but the authors don't **clarify the mechanism of insertion**.

It is a non-specific band in Fig. 3f, which is also present in control samples.

Reviewers' Comments:

Reviewer #1:

Remarks to the Author:

The authors have satisfactorily addressed my comments in this revised manuscript. Although I agree with Reviewer 2 that the improvement over GRAND is relatively modest, I feel the novel approach proposed by the authors has some merit and would be of significant interest to the scientific community.

Reviewer #2:

Remarks to the Author:

My role in evaluating this manuscript was focussed on the Fah animal model. The authors have addressed my concerns adequately. The optimized in vivo correction frequency (without selection) was about 0.8% (Figure 5F).

Reviewer #3:

Remarks to the Author:

The authors have demonstrated good effort in addressing the revision points and carrying out the experiments recommended by the reviewers. As a result, the manuscript has undergone major improvements, significantly strengthening the overall quality and impact of the study. Specifically, the additional experiments elucidate that the mechanism underlying TJ-PE is similar to TPRT, and while this method may not surpass GRAND in efficacy, it presents a viable alternative for employing Prime Editing to promote large insertions.

General Response to Reviewers

We thank the reviewers for their constructive feedback.

Reviewer #1 (Remarks to the Author):

The authors have satisfactorily addressed my comments in this revised manuscript. Although I agree with Reviewer 2 that the improvement over GRAND is relatively modest, I feel the novel approach proposed by the authors has some merit and would be of significant interest to the scientific community.

Our results showed that TJ-PE and GRAND editing mediate similar insertion rates. TJ-PE uses only one long pegRNA and one sgRNA, while GRAND editing employs a pair of long pegRNAs (**Supplementary Fig. 6a**). This unique feature of TJ-PE simplifies the synthesis of pegRNAs for long insertions.

Reviewer #2 (Remarks to the Author):

My role in evaluating this manuscript was focussed on the Fah animal model. The authors have addressed my concerns adequately. The optimized in vivo correction frequency (without selection) was about 0.8% (Figure 5F).

We thank the reviewer for the positive feedback.

Reviewer #3 (Remarks to the Author):

The authors have demonstrated good effort in addressing the revision points and carrying out the experiments recommended by the reviewers. As result, the manuscript has undergone major improvements, significantly strengthening the overall quality and impact of the study. Specifically, the additional experiments elucidate that the mechanism underlying TJ-PE is similar to TPRT, and while this method may not surpass GRAND in efficacy, it presents a viable alternative for employing Prime Editing to promote large insertions.

We thank the reviewer for the positive feedback and constructive comments.